# Mechanical, Leaching, and Microstructure Properties of Mine Waste Rock Reinforced and Stabilised with Waste Oyster Shell for Road Subgrade Use

**DOI:** 10.3390/ma15082916

**Published:** 2022-04-15

**Authors:** Nadia N. Wurie, Junjie Zheng, Abdoul Fatah Traore

**Affiliations:** 1Institute of Geotechnical and Underground Engineering, School of Civil and Hydraulic Engineering, Huazhong University of Science and Technology, Wuhan 430074, China; zhengjj@hust.edu.cn (J.Z.); abdoulfatah@hust.edu.cn (A.F.T.); 2Department of Civil Engineering, Faculty of Engineering, Fourah Bay College, University of Sierra Leone, Freetown 00232, Sierra Leone

**Keywords:** mine waste rock, cement, oyster shell, stabilisation/solidification, strength, heavy metal leachability

## Abstract

Two waste materials, oyster shell (NCOS; non-calcined oyster shell as coarse aggregate and COSP; calcined oyster shell powder as total and partial cement replacement) are used to reinforce and stabilise poorly graded and heavy metal-contaminated mine waste rock (MWR) for pavement subgrade use. Mechanical, leaching, and microstructural tests and analysis were performed on reinforced and stabilised samples to evaluate the effectiveness of the reinforcement and stabilisation of the MWR. Experimental results revealed NCOS and COSP improved the mechanical, leaching, and microstructural properties of the stabilised composite, with a 5% cement–15% COSP–15% NCOS mix being optimal when compared to the control mixes of cement only and no- NCOS. Higher COSP contents beyond 10% reduced the heavy metal contents significantly, but with relatively lower unconfined compressive strengths. Microstructural test results revealed the formation of calcium silicate hydrate (CSH), calcium aluminium silicate hydrate (CASH), ettringite, and calcite as the stabilisation products. Heavy metal complexes in both the cement-only and cement–NCOS–COSP mixes were also found. It is concluded that NCOS reinforced and improved the grading of poorly graded MWR, and that COSP stabilised and immobilised heavy metals present in MWR, thereby improving strength and other engineering properties for subgrade use.

## 1. Introduction

Traditional construction materials, such as ordinary Portland cement (OPC), lime, aggregates, borrow pit gravel, etc., are exhaustible and becoming scarce and/or expensive globally due to the increasing demands of civil engineering [1,2]. Citing [1,2], the sourcing, processing, and manufacturing of these materials is also associated with huge environmental and ecological damage. It is, therefore, imperative for civil and highway engineers to find free/cheap and sustainable alternatives to these traditional construction materials. Over time, many alternatives to cement, lime, aggregates, and borrow pit gravel have been used that have produced similar or improved results when compared to their traditional counterparts, as highlighted by several authors [3,4]. Most of these alternative construction materials are intrinsically waste materials from other industries, and as such, they pose huge pollution and disposal problems worldwide. Two such alternative materials are waste oyster shells (WOS) and mine waste rock (MWR). WOS are the hard exoskeletons of oysters, made predominantly of calcium carbonate (CaCO_3_), and when calcined (burnt), they can yield up to 98% calcium oxide (CaO), as seen in the results of [5]. Refs. [6,7] assert that CaCO_3_ and CaO from WOS are similar to traditional limestone and lime, respectively. This property qualifies WOS to be used in almost every industry. The studies of [8,9,10,11,12,13,14] confirm the use of various forms of WOS in the construction, mining, and water treatment industries. Also, MWR is a non-liquid waste generated from mining activities, which, according to [15], is rock-mined, contains low concentrations of target metals, but are highly contaminated with non-target metals; according to [16], these are harmful to humans, animals, plants, and the environment. This limits its use to mostly mine-site reconstruction and rehabilitation, as seen in [17]. Nevertheless, if these heavy metals present in MWR are stabilised and immobilised with their concentrations reduced to zero (0), or to below specified contamination levels, MWR can be favourably put to far more use, being a granular material.

In road and highway construction, inadequate and unsuitable soils (highly plastic and expansive clays, peats, etc.) are sometimes encountered as subgrade materials. It is advisable to excavate such soils and replace them with suitable borrow pit gravel, as using such in situ soils will require very costly measures to improve their engineering properties. Therefore, MWR, if processed, becomes a viable alternative for use in road pavement works. Furthermore, calcined oyster shell powder (COSP) was successfully used in [18,19,20,21,22] as a supplementary cementitious material (SCM) for total or partial cement replacement to stabilise and immobilise heavy metals found in mine tailings and firing range soils, and was adopted in [14] to stabilise and improve lateritic soil strength. Crushed non-calcined oyster shell (NCOS) was also used in [23] to reinforce and improve the grading of poorly graded subgrade soil. It is, therefore, hypothesized that the appropriate size of NCOS will reinforce and improve the grading of poorly graded MWR, whilst COSP will stabilise and immobilise the heavy metals present in MWR, thereby improving its strength and other engineering properties for road subgrade use.

Apparently, NCOS and COSP have never been used to reinforce and stabilise MWR for use in roadworks. Therefore, the objective of this research is to study the mechanical, leaching, and microstructural properties of MWR reinforced and stabilised with NCOS as a coarse aggregate and COSP as a partial and total cement replacement for use as a road subgrade material. Zinc (Zn), copper (Cu), and lead (Pb) are the selected target heavy metals that need to be stabilised and immobilised. This is because they are encountered the most in contaminated soils and mining sites worldwide [24]. Moreover, all three are amongst the first five heavy metals in industrial metal production [24]. Kumpiene et al. [18] affirms that these three elements also have similar responses to the same amendments. Unconfined compressive strength (UCS) and toxicity characteristic leaching procedure (TCLP) tests will be performed on cured samples of different mixes, and the results compared to results of the control mixes. Results will also be checked against strength and leachability specifications for materials that are used for road subgrades. Mineralogical and microstructural characteristics of the stabilised composites will also be investigated by x-ray diffraction (XRD) and scanning electron microscopy (SEM) in order to better understand the mechanisms and products of stabilisation that caused improved soil properties. The combined use of MWR, NCOS, and COSP translates to a two-waste use composite that is proven experimentally to offer a free/cheap alternative to traditional road construction materials, resulting in reduced civil engineering project costs whilst addressing environmental challenges. This is a win–win for project execution cost and environmental/ecological conservation and preservation. In addition, the synergetic effect of such a combination could yield improvements in the strength and performance of road subgrades replaced with such waste materials. It is worth noting that comparison or reference will be made regarding the use of cement, lime, NCOS, and COSP in waste treatment and civil works, since extensive research and data is available for them, rather than for road pavement research.

## 2. Materials and Methods

### 2.1. Materials

Ordinary Portland cement (OPC—Type I) grade 42.5, MWR, NCOS, and COSP were used in this research. MWR (particle size ≤ 5 mm, to simulate poor subgrade soil) was obtained from Huangshi Copper Mine, Hubei, China. Huaxin OPC grade 42.5 was bought from Jincheng Yangzhou Building Materials Technology, Wuhan, Hubei, China. NCOS and COSP were procured from a local supplier in Wuhan. NCOS (10–13 mm), COSP, and MWR (≤5 mm) are shown in Figure 1. The particle size distribution (PSD) of unstabilised/unreinforced MWR (≤5 mm) and NCOS (10–13 mm) and the lower and upper limits for suitable subgrade material are shown in Figure 2. The basic physical properties and chemical compositions of the materials used were determined according to the China Standard GB/T 50123-2019: “Standard for Geotechnical Testing” and are as given in Table 1. The pH was measured according to ASTM D4972-18. Chemical compositions were determined via x-ray fluorescence refractometry (XRF).

#### Optimal Material Parameters for Civil and Waste Treatment Works (Using Binders and Seashells)

Ten mixes were designed and are as shown in Table 2. Throughout this research, the various mixes will be referred to as, for example, “Mix 1A” or “M1A”, etc. NCOS was included in the mix design to improve the grading properties of the MWR, since it is deficient in coarse-sized particles, in line with Ruiz and Farfán [25]. Past literature suggests an optimal size for crushed oyster shell (OS), when used as a coarse aggregate replacement in concrete mixes, of 10–13 mm, as observed in the work of [10]. Ruiz and Farfán [25] considered limiting crushed Peruvian scallop shells’ use in subgrade stabilisation to 20%, because of acceptable limits of elongated particles in the subgrade. In addition, the review completed by Eziefula et al. [26] cited researchers who, for workability and strength reasons, in concrete, limited the replacement of both fine and coarse aggregates with NCOS to 10–12%. Thus, 15% by weight, 10–13 mm NCOS was used in five of the ten mixes. Most literature had an optimal binder content for poor subgrade stabilisation and improvement between 3–10%, depending on the soil type and the type of binder used, as suggested by authors [5,14]. Refs. [20,22,27], in context, adopted, for heavy metal stabilisation, optimal binder contents between 5–15%. According to Kumpiene et al. [18], the remediation of soils contaminated with several metals at different concentrations requires effective solutions for all target elements. Therefore, higher binder contents are needed to successfully stabilise all heavy metals. Higher binder contents provide more sorption sites for all contaminants present. Multiple-heavy-metal-contaminated soils also usually have lower pHs (more acidic) than their singly contaminated counterparts; thus, there is a need for higher alkaline binder contents to neutralise the acidity of the soil, as suggested by [28,29]. Twenty percent (20%) binder content was thus chosen for this study, as this is conservative for both the reinforcement and stabilisation of the MWR. The chosen percentages of NCOS and COSP extends previous works and seeks to evaluate the final effect.

### 2.2. Methods: Mechanism of Stabilisation and Immobilisation, Tests and Experiments

#### 2.2.1. Mechanism of Stabilisation and Immobilisation of Heavy Metal Contaminated Soils

Several authors [4,30,31] define soil stabilisation as the process of chemically or physically altering the properties of problematic soils in order to enhance their engineering qualities for use in engineering and infrastructural applications. This, according to [1,32,33,34,35,36,37], involves the addition of cement, lime, and other industrial by-products. Mainly, the objective is to increase bearing capacity, durability, and resistance to weathering, and to decrease swell/shrink potential (volume stability), permeability, to reduce the plasticity of soil, and to reduce pavement thickness (in the case of subgrades), according to [30,31,38]. Refs. [30,36,38,39,40] identified soil type, stabiliser type, stabiliser content, stabiliser test procedures, water content, mixing, compaction, curing time, curing conditions (temperature and humidity), gradation, and pulverization as the many factors that affect soil stabilisation. Cement and lime have been used for stabilisation, in isolation or together, in the studies of [32,33,36,41,42,43,44]. When water is present and sufficient cement added, hydration products are formed, and these products induce cementation between the soil particles, subsequently causing increased shear strength, according to A. Athanasopoulou [45]. Also, when lime (in the presence or absence of cement) is used to stabilise soils, reactions between soil and lime in the presence of water goes through four basic mechanisms: (i) cation exchange or ion exchange; (ii) flocculation and agglomeration; (iii) pozzolanic reaction; and (iv) carbonation. According to authors [46,47,48], all of these cause improved material properties in the soil. The mechanisms of chemical stabilisation of heavy metals in soils is almost similar to the general stabilisation of soils, i.e., cation exchange, flocculation and agglomeration, cementitious hydration, and pozzolanic reaction.

However, some form of physical solidification of the soil composite almost always accompanies heavy metal stabilisation of soils, hence the term “Stabilisation/Solidification” (S/S) by Shi et al. [49]. S/S is the chemical fixation and physical encapsulation of contaminants within the matrix of a cementitious binder added to contaminated soil or hazardous waste to make it more chemically stable and environmentally acceptable as waste, thus limiting the leaching of contaminants [13,18,28]. S/S processes are currently being used to immobilize Pb^2+^ and Cu^2+^ in military firing range soils, and also to immobilise heavy metals in contaminated soils and sludges, as in [20]. Many types of S/S agents are available, such as cement, lime, fly ash, cement kiln dust, etc., and these have been used in S/S works, according to several authors [29,50,51,52,53,54,55,56,57]. Natural lime wastes, such as cockle and oyster shells (mostly in their calcined form), have also been used in [18,21,58]. Waste oyster shell (WOS), rich in CaCO_3_ and CaO, serves as a liming material for stabilising heavy metal-contaminated soils due to the formation of insoluble metal hydroxides at alkaline pHs [13,21]. Solidification of the polluted material restricts its contact with air and groundwater, as suggests Burlakovs et al. [59]. Amendments used to immobilise contaminants decrease heavy metal leachability, hydraulic conductivity, and their bioavailability by inducing various sorption processes, e.g., the formation of stable complexes with organic ligands, adsorption to mineral surfaces, surface precipitation, and ion exchange, according to [18,60]. Precipitation as salts and co-precipitation can also contribute to reducing the mobility of contaminants, as seen in [61,62]. Kumpiene et al. [18] state that many factors influence these different sorption/dissolution processes, such as the type of soil constituents, the redox potential, the pH, the cation exchange capacity, etc., and a single mechanism rarely accounts for the immobilisation of heavy metals in soil. One of the most important factors is pH, as it has a very strong impact on heavy metal mobility in soils. Chemical precipitation occurs at high-alkaline pHs, usually between pH 9–11. In general, the solubility of heavy metals (except for molybdenum, Mo) can generally be reduced under high-pH conditions, causing a reduction in the metal phytoavailability of contaminated soil [63]. The presence of divalent iron (contained in most industrial wastes) also speeds up such immobilisation, explains R. B. Kogbara [27]. In summary, cementitious binders react with metals and cause the formation of hydroxides, carbonates, and silicates, etc., with little or no solubility, explains Burlakovs et al. [59]. Authors [4,29,45,59] also point out that S/S usually leads to the increased compressive strength and decreased compressibility of the treated material. Edalat-Behbahani et al. [64] also explain that S/S composites have properties similar to those of concrete or rock.

#### 2.2.2. Preliminary Tests and Sample Preparation

The original MWR was contaminated with very low levels of the target metals, i.e., Zn, Cu, and Pb. It was, therefore, spiked above contaminated levels with Zn(NO_3_)_2_·6H_2_O, Cu (NO_3_)_2_·3H_2_O, and Pb (NO_3_)_2_, obtained from Wuhan Xinshenshi Chemical Technology Co., Ltd., Wuhan, Hubei, China. The metal nitrates were used for soil spiking because nitrate is inert to cement hydration and, consequently, does not adversely affect the engineering properties of the stabilised soil, according to Du et al. [65]. The Chinese EPA [66] states that zinc (Zn), copper (Cu), and lead (Pb) concentrations in soils should satisfy the following: Zn ≤ 300 mg/kg or ≤100 mg/L, Cu ≤ 200 mg/kg or ≤15 mg/L, and Pb ≤ 80 mg/kg or ≤5 mg/L. MWR was spiked 15-times above contamination levels with the three (3) heavy metals, i.e., 4500 mg/kg with Zn, 3000 mg/kg with Cu, and 1200 mg/kg with Pb. The initial heavy metal concentrations in the spiked MWR are as follows: Zn = 216 mg/L, Cu = 84 mg/L, and Pb = 13.2 mg/L. This spiked soil was then used for all tests and experiments in this research.

Particle size distribution and UCS test results of the unstabilised MWR in Figure 2 and Table 1, respectively, prove that the unstabilised MWR fails to meet grading and strength specifications for road subgrade material. The PSD for the ten (10) design mixes are as shown in Figure 3. It can be seen that the addition of 10–13 mm NCOS improved the grading of the MWR (curves closer to the grading envelope). This further emphasizes the fact that the addition of NCOS to the mix improved its grading. The PSD of the A samples are almost the same, and those of the B samples are also similar. Samples of each mix were prepared in UCS moulds at their respective optimum moisture contents (OMCs) and maximum dry densities (MDDs). Demoulded samples were wrapped in cling film and cured in a curing chamber at 23 ± 2 °C and 95 ± 5% relative humidity for 7, 14, and 28 days, to be used for strength, leachability, and microstructure tests. All tests/experiments were performed in duplicate and average values were reported only if the individual measurements were within an error of 10%, as suggested by [21,22]. The results showed that the values of the standard deviation were less than 5% for the duplicates, demonstrating the reproducibility of the test results.

#### 2.2.3. Unconfined Compressive Strength (UCS) Test

As Aldeeky et al. [67] explain, UCS is used to assess the shear strength parameters of soil and cementitious mixes (especially for subgrade use), and Burlakovs et al. [59] mention that it is also an important parameter used in determining the effectiveness of the stabilisation and solidification (S/S) of heavy metal-contaminated soils. Unsoaked and soaked UCS tests were performed according to ASTM D 1633—17 and, for the reasons given therein, with the UCSs converted and given in MPa. Statistical analysis using Minitab 21.1.0.0 (x64) one-way ANOVA was also undertaken to assess the performance of the different mixes, whereby *p* ≤ 0.05 was considered statistically significant.

#### 2.2.4. Toxicity Characteristic Leaching Procedure (TCLP)

The leachability of heavy metals from the cured stabilised soil samples was determined using the toxicity characteristics leaching procedure (TCLP), and the concentrations of heavy metals (Zn, Cu, and Pb) were measured using “flame atomic absorption spectrometry”. Crushed UCS samples at the various curing ages were sieved, and particle sizes ≤ 2 mm were used for the TCLP, which was performed according to EPA TCLP method 1311 (US EPA, 1992) [68]. Quoting Islam et al. [69], TCLP evaluates the environmental performance of the stabilised sample, i.e., the capability of the various mixes in retaining heavy metals. According to R. B. Kogbara [27], leachability is mainly pH-controlled.

#### 2.2.5. X-ray Diffractometry (XRD) and Scanning Electron Microscopy (SEM)

X-ray diffraction (XRD) analyses and scanning electron microscopy (SEM) were carried out, respectively, to study the crystallization characteristics and morphology of the S/S soil samples. These explain the mineralogical and chemical changes and the subsequent microstructural modifications that take place in stabilised soil as a result of stabilisation, as stated by authors [14,69]. Crushed 28-day UCS samples were used for both XRD and SEM analysis. For XRD analysis, samples were sieved through a 0.075-mm sieve to obtain a fine powder, and were scanned in ranges from 0° to 90° (2θ) using a Rigaku D/Max-2500 X-ray diffractometer with a Ni-filtered Cu-Kα radiation source at 40 kV and 30 mA, and with 0.02° intervals and a counting time of 1 s per step in order to identify the crystalline phases. Jade software version 7.1 (MDI 2005) [70] was used to conduct the qualitative analyses of the XRD patterns, using the patterns present in the International Centre for Diffraction Data database (ICDD 2002) [71] as reference. For SEM analysis, SEM SU8010, Hitachi, Japan, equipped with an energy dispersive X-ray (EDX) spectroscope, was employed to acquire micrographs at different resolutions and magnifications, as well as the chemical compositions of the stabilised samples. Samples were placed on aluminium stubs using double-sided carbon tape prior to SEM analysis, and were then placed in the machine for image capture at 5 kV.

## 3. Results and Discussions

### 3.1. Heavy Metal Leaching Behaviour of Stabilised MWR

#### 3.1.1. Curing and Leachate pH

Results are as shown in Figure 4 and Figure 5. The same trends were observed for the 7-, 14-, and 28-day pHs after curing and leaching both sample A mixes (with NCOS) and sample B mixes (no-NCOS). For pH after curing and leaching, the pHs of the A samples are larger than those of the B samples because the added NCOS in the A samples increases calcium, which increases pH. Calcium from both COSP and NCOS is available to mix with water to give Ca(OH)_2_, which yields higher pHs in the A samples. Less calcium is available in the B samples (no-NCOS); thus, these have relatively lower pHs after curing and leaching, compared to the A samples. For both the A and B samples, the pHs after leaching are lower than those after curing. This is due to the rigorous and aggressive acidic leaching conditions. The high pH after curing is reduced due to the acidic leaching conditions; the high pH after curing buffers and neutralises the acidic conditions during leaching, lowering the pHs during leaching. Additionally, due to the acidic leaching environment, already stabilised and immobilised heavy metals are dissolved and remobilised, and this increases the acidity of the soil composites, thus lowering the pHs after leaching.

For both the A and B samples, as the cement content decreases and the COSP (which is physically and chemically almost the same as lime) content increases, the pH after curing and leaching increases. This is due to the increase in calcium present in COSP, which raises the pH as the COSP content increases; this trend is also reported by Ok et al. [13]. This observation is appreciated, as high pHs will act as a buffer in acidic environments and will create an enabling environment for insoluble metal precipitates to be formed and maintained in the stabilised composite. Mix 1A and 1B are the control mixes, against which the other mixes are compared.

Comparing the 7-, 14-, and 28-day pH results revealed that, after curing, the 7-day pH > 14-day pH > 28-day pH. This is due to the fact that, at early stages of curing, not all hydrolysed calcium has been used to neutralise the acidity of the heavy metals—thus, there are high pHs at 7 days. A lot of free Ca^2+^ ions in solution have still not yet been used to form cementitious hydration and pozzolanic reaction products, as these products are mostly formed after 7 days of curing. As curing age increases, more available Ca^2+^ reacts with silica (Si) and alumina (Al) in cement and soil to form cementitious hydration and pozzolanic reaction products, or reacts with atmospheric Co_2_ to form carbonation products (i.e., calcium silicate hydrate (CSH), calcium aluminate hydrate (CAH), calcium aluminium silicate hydrate (CASH), calcite, etc.); hence, there are less calcium ions and, consequently, lower pHs at the latter days. Thus, at 28 days, more calcium would have been used to form stabilisation products, accounting for the lower pHs at 28 days.

On the other hand, after leaching, the 7-day pHs < 14-day pHs < 28-day pHs. This is due to the fact that at 7 days, many stabilisation products have not been formed, so as soon as the soil becomes acidic, the not-too-stabilised or immobilised heavy metals go into solution and lower the pH of the soil, hence the low leachate pHs of the 7-day cured samples. As already stated, heavy metals present in soils cause soil pH to be low. Heavy metals at 7 days are in the preliminary stages of stabilisation; therefore, there are low pHs, but as the curing age increases, and as more stabilisation products are formed, these heavy metals are stabilised and immobilised, consequently leading to relatively high leachate pHs at 28 days. As stated earlier, cementitious hydration, pozzolanic reactions, and carbonation products stabilise and immobilise the heavy metals in the soil. At the 28-day curing age, the heavy metals are already stabilised and immobilised, so even after rigorous and aggressive acidic leaching conditions, when the soil becomes acidic, the heavy metals hardly dissolve, subsequently resulting in the relatively higher leachate pHs, as compared to those at 7 days. All these recorded observations are in agreement with the results of [72].

#### 3.1.2. Heavy Metal Leachate Concentration

Results are as shown in Figure 6a–f. The A samples’ heavy metal concentrations are less than the B samples’ heavy metal concentrations. This is due to the fact that additional calcium from NCOS increases the pH and, in turn, dissolves Si and Al, which, in turn, causes more CSH, CASH, etc., to be formed. These cementitious hydrates encapsulate the heavy metals and immobilise them, as will be seen in the SEM micrographs, hence their lower concentrations.

As cement content decreases and COSP content increases, heavy metal leachate concentrations decrease. Heavy metal stabilisation and immobilisation is promoted by a high pH, which the increasing COSP content provides.

Moreover, the 7-day concentrations > 14-day concentrations > 28-day concentrations (28 days being the lowest). This is because, as mentioned earlier, at 7 days, hydration products are not well formed and cannot stabilise and immobilise the heavy metals in the soil; therefore, their concentrations remain high in the soil. At 28 days, however, stabilisation reactions are well advanced, and the stabilisation products are properly formed, and these stabilise and immobilise the heavy metals in the soil; thus, there is a lower heavy metal concentration at 28 days.

It can be seen from the TCLP results that enough binder, i.e., cement and/or COSP, was available to provide sorption sites for the precipitation of insoluble metal complexes, as at the 28-day curing age, all concentrations for Zn, Cu, and Pb for the various mixes were almost zero (0) or below contamination levels, as given by the Chinese EPA (Zn ≤ 100 mg/L, Cu ≤ 15 mg/L, and Pb ≤ 5 mg/L) [66]. Refs. [19,20] also had similar results in their studies. This shows that the combination of OPC, COSP, and NCOS was effective at simultaneously stabilising and immobilising all heavy metals present in the MWR. The synergistic effect of the multi-contamination of the MWR by the three (3) heavy metals did not affect the stabilisation and immobilisation of each one in the presence of the other.

From the TCLP results, M4A, M4B, and M5A, M5B, with little or no cement, also had the lowest heavy metal concentrations after leaching, but relatively low UCSs. This implies that the high alkaline environment that calcium in COSP created played a great role in the stabilisation and immobilisation processes. In the presence of little or no cement, it can be seen that COSP effectively stabilised and immobilised the heavy metals in the MWR. This comes as no surprise, as [18,20], in their research, confirmed that lime can effectively decrease the mobility of Pb and Cu in heavy metal-contaminated soils by increasing soil pH without necessarily causing strength gain. These results further confirm the report of [73], that lime on its own lacks the ability to form adequate hydration products to the effect of considerable strength gain. In these mixes, heavy metal concentrations after leaching were the lowest, implying COSP on its own successfully stabilised and immobilised all three (3) heavy metals, but could not produce enough stabilisation products to affect UCS considerably. COSP needs cement to provide Si and Al, in addition to those found in soil, to form cementitious hydrates (pozzolanic reaction products) which harden and increase UCS. In such mixes with little or no cement, strength gain mostly comes from the Si and Al present in the soil combining with the Ca^2+^ from COSP to form CSH, CASH, etc. Moreover, much of the UCS in such mixes comes from carbonation products (calcite), as will be later seen from the XRD and SEM results.

At the 28-day curing period, the heavy metals in the various mixes were stabilised at pHs between 11.7–12.42 for the A samples, and between 11.67–12.34 for the B samples. Extraction pHs for the A samples were between 6.29–11.95, and for the B samples they were between 6.1–11.83. These pHs are similar to the pH results in [21,22]. This is good, as such pHs will maintain alkaline conditions that will ensure the insoluble metal complexes remain insoluble. The conditions under which TCLP was performed were really rigorous, aggressive, and extreme, and such conditions rarely occur in nature, but even under such conditions in this study, heavy metal concentrations were almost zero (0), or below contamination levels. The S/S procedure can, thus, be considered successful and permanent.

### 3.2. Unconfined Compressive Strength (UCS)

#### 3.2.1. Unsoaked UCS of Stabilised MWR

Results are as shown in Figure 7. For any particular mix, sample A UCSs are greater than sample B UCSs. This is because the 10–13-mm NCOS added to sample A mixes improved its grading, and during the curing and subsequent hydration of the binders, this NCOS created bone structures that improved and strengthened the connections between soil particles and hydration products, thus yielding a stronger composite. Consequently, the A samples had relatively higher UCSs than their corresponding B counterparts. NCOS is also coarser and angular, increasing the occurrence of interlocking within the composite, and yielding a stronger skeleton, hence the greater amount of UCS. This was also noted in the results of [74].

For any particular mix, 28-day UCSs > 14-day UCSs > 7-day UCSs. This is because UCS increases with curing age, and most of the mechanical strength is usually developed around or after 28 days.

As cement content decreases and COSP content increases, UCS increases for mixes 1A, 2A, and 3A, and decreases for mixes 4A and 5A. This is owing to the fact that, in mixes 1A, 2A, and 3A, the cement content is greater than or equal to the lime content. According to [45], cement hydrates and forms strong complexes, but lime does not. Refs. [31,73] pointed out this limitation of lime in cementitious mixes. Nevertheless, even when cement content decreases, lime provides more calcium, which reacts with Si and Al in both cement and soil to form CSH, CASH, etc., which are cementitious hydrates that bind soil particles together strongly, hence the increase in UCS. It should also be noted, according to [75,76], that considerable shrinkage occurs in cement-only mixes; this lowers UCS, and so even though it is expected for cement-only mixes to yield greater UCS than cement–lime mixes, cement-only mixes will have relatively lower UCS than cement–lime mixes. Less shrinkage occurs in mixes with COSP, as COSP is finer than cement and fills micro-pores, yielding a refined pore structure, as is shown in Table 1 and alluded to by Lertwattanaruk et al. [77]. Zn also retards cement hydration, consistent with findings previously reported by Lo et al. [78]. All the above reasons explain why M2 and M3 mixes have higher UCSs than M1 mixes.

In mixes 4A and 5A, UCS decreases because there is little or no cement to form sufficient cementitious hydration products. As stated earlier, lime on its own does not possess hydration properties. Soil–lime mixes or soil–lime with little cement only gains strength through pozzolanic reaction products, carbonation products, or hydration products enhanced by whatever little amount of cement is available. Such high COSP contents also cause disturbances in the soil skeleton, consequently decreasing cohesion; hence, the UCS is lowered. Authors [45,79] mentioned the foregoing phenomenon in their studies. Lime on its own has neither appreciable cohesion nor friction. Lime, therefore, needs cement to complement its strength gain. Mix 4A, even with less cement and more COSP, however, gave a 28-day UCS that was higher than that specified for road subgrade use, that is, 9 MPa > 1.5 MPa, as specified in [80]. It can even be used as a sub-base material [80]. This will decrease pavement layer thickness and subsequently decrease construction costs, thus making Mix 4A economical. All these recorded trends and observations are in agreement with the results of [4,33,45].

Statistical analysis using Minitab 21.1.0.0 (x64) one-way ANOVA confirmed that the results were statistically significant for all curing periods, with *p* values < 0.05 (*p* = 0.000), further confirming the results obtained.

#### 3.2.2. Soaked 28-Day UCS of Stabilised MWR

Slight adjustments were made to ASTM D 1633—17 for this test, in which the 28-day samples were used and soaking was conducted for 48 h. Such a long saturation of stabilised soil samples with water before UCS testing simulates the worst-case conditions on site, as saturation weakens and destroys some of the compounds formed by the stabilisation process. According to Etim et al. [14], such a test is necessary to extend the knowledge of the mechanical behaviour of stabilised soil in a wet environment. From the unsoaked 28-day UCS and the soaked 28-day UCS test results and plot in Figure 8, the soaked UCSs followed the same trend as their corresponding unsoaked counterparts, but it can be seen that soaking the samples for forty-eight (48) hours caused the UCS to decrease slightly. Etim et al. [14] reported similar trends between unsoaked and soaked UCS results.

### 3.3. XRD Analysis of Stabilised MWR

From the leachability and UCS test results, M4A can be regarded as the most economical and viable mix, as it satisfies the respective specified standards, uses the least cement content, and has a high OS content. Therefore, XRD and SEM tests were performed on crushed 28-day samples of M4A and M4B, with the two control mixes (M1A and M1B) for comparison purposes. Data between ranges 10° and 70° (2θ) were recorded as significant peaks were detected between this range; no peaks were detected below 10° or above 70°.

The 28-day XRD patterns of M1A, M1B, M4A, and M4B are shown in Figure 9, with quartz being the predominant compound in all mixes, reflecting the nature of MWR. Quartz peaked at a Bragg angle of 2θ = 29°, with decreasing intensities from Mix 1 to Mix 4, as the calcium content of the mixes increased. This decrease in intensities of quartz occurs as more soil minerals react with increased calcium from COSP to form stabilisation products. The M1A and M1B samples had more quartz than cementitious hydration products, whilst the M4A and M4B samples had less quartz and more stabilisation products (due to the high COSP content). Mix 1B had more quartz than the other three (3) mixes, as it had the lowest calcium content. For M1A (all cement plus NCOS) and M1B (all cement), the main stabilisation products are CSH and CASH, with some calcite. For M4A (5% OPC–15% COSP–15% NCOS) and M4B (5% OPC–15% COSP), the main stabilisation products are CSH, CASH, and ettringite, with some more calcite. All these are typical of calcium-based stabilisers; this is consistent with previous results from [22,69,81]. This is also expected, as from the chemical composition of the materials in Table 1, cement provides added Si and Al but not as much calcium as COSP and NCOS. Therefore, there are more cementitious hydration products in Mix 1 than in Mix 4, and more pozzolanic reaction and carbonation products in Mix 4 than in Mix 1. Ettringite and calcite are the major differences between Mix 1 and Mix 4. The presence and quantity of ettringite and calcite in M4 is expected, due to their high calcium content (COSP) reacting more with OPC and soil minerals to form stabilisation products, and due to calcium in solution reacting with atmospheric CO_2_ to form calcite, respectively. These results are in conformity with [21]. Ref. [45] stated that, at high lime (COSP) contents, increased carbonation of lime occurs. Therefore, it is evident that the high COSP and NCOS in Mix 4 enhanced the formation of more carbonation products and also adequate stabilisation products. XRD patterns displayed broad peaks of calcite at 2θ = 34°, similar to ranges of that reported by [44]. Hydrocalumite was detected as a minor hydration product early on, during the curing process. Portlandite (Ca(OH)_2_, a major hydration product, was not detected in M1 (all cement mixes). This is likely due to Zn’s retarding effect on cement hydration, consistent with findings previously reported by [78]. However, some portlandite was detected in M4, owing to the high calcium content (COSP) being able to reduce the retardation of cement hydration caused by Zn. This can also be due to the fact that when calcium content is small, portlandite is mostly consumed by natural carbonation (yielding calcite), but at high COSP content, both portlandite and calcite coexist, as is similarly reported by [5].

The high alkaline pHs in all four mixes caused the precipitation of insoluble metal complexes in varying proportions, consequently causing heavy metal retention. Peaks of the insoluble metal complexes mostly overlapped with those of the stabilisation products. Peaks of zinc oxide and copper oxide were detected in all mixes, indicating that Zn and Cu were mainly precipitated as oxides. Ref. [82] reported similar results for zinc retention, i.e., Zn is usually bonded to carbonate and iron oxide phases. Zn and Pb were also stabilised/solidified on the surface of CSH by an adsorption mechanism and chemical reactions to form insoluble zinc and lead silicates, respectively. The Zn tetrahedral can be bonded to CSH tetrahedral silicate chains, causing Zn retention, according to [82]. Calcium zincate (CaZn_2_ (OH)_6_·2H_2_O) was also detected in all cases; this is a result of the reaction between Zn and Ca(OH)_2_. In M4 mixes, zinc was absorbed by portlandite and precipitated as zinc hydroxide. Insoluble metal complexes in M1A and M1B were mostly incorporated in CSH and CASH, forming silicates, and those in M4A and M4B were mostly incorporated in calcite, ettringite, and portlandite, forming oxides, hydroxides, etc. These differences affected the UCS, TCLP, and SEM–EDX (seen later) results of the different mixes. It can be seen from Figure 9 that M1 with more OPC had more CSH and CASH, which are voluminous hydration products yielding more strength (according to [83]), accounting for the relatively higher UCS of M1 as compared to M4. However, M1′s heavy metal concentrations were relatively high, due to its relatively lower pH, whilst M4 with less OPC and more COSP had relatively less CSH and CASH and, consequently, a lower UCS, but lower heavy metal concentrations due to its high pH. The high COSP content increased its pH, which caused more pozzolanic and carbonation products to be formed, and which further immobilised the heavy metals and lowered their concentrations within the stabilised composite. SEM–EDX results also confirm all of the foregoing.

M1A and M4A products had greater intensities than their corresponding B counterparts, implying more products and greater stabilisation and strength. This is due to the added NCOS, which provided additional calcium for more stabilisation products to be formed. It can be concluded from all the stabilisation products that Zn, Cu, and Pb were stabilised and immobilised by adsorption and incorporation into CSH, CASH, ettringite, and calcite, and precipitated as insoluble silicates, carbonates, oxides, and hydroxides. These simultaneously caused heavy metal retention and decreased their concentrations in the soil composites. Previous studies by several authors [19,21,22,31,69,82,84,85,86] further reinforce the concluding observations.

### 3.4. SEM Analysis of Stabilised MWR

The SEM micrographs of the crushed 28-day cured samples of M1A, M1B, M4A, and M4B are as presented in Figure 10. SEM–EDX spectrographs and elemental atomic percentages are shown in Figure 11 and Table 3, respectively. SEM–EDX results indicate that effective Zn, Cu, and Pb immobilisation using the combined OPC–COSP–NCOS treatment is strongly associated with the hydration and pozzolanic reaction products (CSH and CASH), ettringite, and the carbonation product (calcite). This further supports the findings in the XRD results. These stabilisation products filled pores and cemented MWR particles, causing strength development and increased UCS. Insoluble heavy metal complexes are also precipitated within these stabilisation products, stabilising and immobilising the heavy metals in the soil. The results of [14] also depict and reinforce a similar trend. The results of [87] reported some ettringite close to, or in the location of, seashell powder; this might partially explain why seashell powder enriches the hydrated composite and promotes the precipitation of hydration products, as it serves as a nucleation site. Moreover, according to [86], ettringite has a needle morphology with diverse divalent cations such as Zn^2+^, Cu^2+^, Pb^2+^, etc., replacing Ca^2+^ in the ettringite minerals. This indicates that Zn, Cu, and Pb are incorporated into the ettringite crystal structure. Ref. [88] also reported Zn incorporation within CSH gels from their SEM–EDX test results. XRD results from this study confirms this also. All of these findings validate the results of this study. The morphological evolution of all the stabilisation products in the different mixes, as observed in the SEM micrographs, were also identified in the XRD analysis.

Needle-like ettringite, gel-like CSH, platy CASH, and bulky calcite were highly visible in all four micrographs; these are typical of calcium-based stabilizers and are also reported in the studies of several authors [21,88,89,90]. However, differences in the intensities and quantities of these products can visibly be seen in Figure 10a–d, reflecting the differences between the four mixes. M1A and M1B, with more cement and relatively less calcium, have a less spongy look, with M1B being the least spongy (no NCOS). M4A and M4B, with more calcium, had a spongier look, with M4B being a little less spongy (no NCOS). The presence and fineness of COSP and the precipitation of calcite further densified the M4 mixes, giving them a more compact structure (less pores are seen in M4A and M4B). Ref. [5] also recorded similar results. The microstructural arrangements in Figure 10c,d showed better immobilization efficiency of Zn, Cu, and Pb against acidic TCLP conditions, as compared to Figure 10a,b. The formation of ettringite and increased calcite in M4 also resulted in improved mechanical strength of the high COSP–NCOS mixes, which would have otherwise had much less UCS. Less reticulated honeycomb CSH can be seen in M1A and M1B; this can be attributed to Zn impeding the cement hydration process by coating the cement grains, as previously pointed out by [91]. However, due to the high cement content of M1A and M1B, they still had considerable UCS, as compared to M4A and M4B. The UCS of M4A and M4B are the combined effects of the pore structure refinement and calcite precipitation induced by the high COSP content, as well as pozzolanic reaction products. This is also confirmed by the results in [5], reflecting mixes with high COSP content. SEM—EDX elemental atomic percentages and spectrographs revealed that M4 mixes had the highest Ca contents, with comparatively lower Si and Al, implying increased alkalinity and more calcites being formed (due to increased carbonation) and comparatively lower CSH and CASH, when compared to the M1 mixes. TCLP, UCS, and XRD results of the M4 mixes are further confirmed, with M4 having increased pH and calcite formation and precipitating more insoluble heavy metal complexes, which increased the stabilisation and immobilization efficiency of the heavy metals, and significantly decreased their concentrations within the stabilised composite. The relatively lower Si and Al in the M4 mixes yielded comparatively lower CSH and CASH; thus, the relatively lower UCS of the M4 mixes, as compared to the M1 mixes, further confirms the UCS results.

### 3.5. Inter-Relationships between Mechanical, Leaching, and Microstructural Properties of Reinforced and Stabilised MWR

In summary, from all the above test results and discussions, it can be seen that, as OS content increases, pH increases and the heavy metal concentration of the stabilised composite decreases. Also, as pH increases and heavy metal concentration decreases, the UCS of mixes 1, 2, and 3 increases (because of the high cement content) and the UCS of mixes 4 and 5 decreases (because of little or no cement). Also, as the pH increases and the heavy metal concentration decreases, the calcite content increases, densifying the composite.

## 4. Conclusions

This study proposed a “two-waste material” composite, reinforced and stabilised/solidified for road pavement subgrade use. COSP, as a total and partial cement replacement, and NCOS, to improve MWR grading, were used as alternative construction materials. A series of tests were conducted to evaluate the effect of NCOS and COSP content on the mechanical, leaching and microstructure properties of the reinforced and stabilised material. The following conclusions were drawn:The presence of NCOS and the increase in COSP content and curing age increased curing and leachate pH, decreased the leachability of Zn, Cu, and Pb, and decreased their concentrations in the stabilized composite to almost zero (0), or below contamination levels. This is due to the high-pH environment ensured by NCOS and COSP, thus stabilising and immobilising these heavy metals;The addition of 15% NCOS affected UCS positively with an evident increase in UCS. Samples without NCOS had relatively lower UCSs. Moreover, as cement content decreased and COSP content increased, UCS increased up to 10% cement replacement with increases in curing age. At COSP contents higher than 10%, UCS decreased slightly. Nevertheless, higher COSP content mixes with little or no cement still yielded UCSs that satisfy specifications for both subgrade and sub-base use. The twenty-eight (28)-day UCS for the A mixes are 1.24-times greater than their corresponding B counterparts. The average 28-day UCSs of M1A, M2A, and M3A are 2.55-times greater than those of M4A and M5A. Also, the average 28-day UCSs of M1B, M2B, and M3B are 2.59-times greater than those of M4B and M5B. Statistical analysis, using Minitab 21.1.0.0 (x64) one-way ANOVA, confirmed that the results were statistically significant for all curing periods, with *p* values < 0.05 (*p* = 0.000);The UCS results of the 28-day samples soaked in water for 48 h were just slightly lower than their corresponding unsoaked counterparts. The 28-day unsoaked UCSs are, on average, 1.33-times greater than their soaked counterparts. Cementitious hydration, pozzolanic reaction and carbonation products, COSP, and NCOS refined the pore structure of the composites by filling pores, densifying the composites and, subsequently, yielding improved mechanical properties. Therefore, even saturation in water did not adversely affect UCSs;In addition to the main hydration and reaction products (CSH, CASH, and ettringite), XRD patterns revealed that NCOS and COSP supplied extra CaO, which facilitated the formation of Calcite. CaO reacts with atmospheric Co_2_ to form calcite. Calcite, an insoluble carbonate with considerable strength, was visibly seen. The increased formation of calcite resulted in the improved mechanical strength of high-COSP–NCOS mixes, which would have otherwise had much lower UCSs. Calcite can also be seen to contribute greatly to higher Zn, Cu, and Pb stabilisation efficiency in M4, as it precipitated these heavy metals as insoluble complexes. The presence of all these hydration, reaction and carbonation products will ensure the permanence of treatment;SEM micrographs revealed denser microstructures for mixes with NCOS and COSP, as a result of the additional formation of calcite, which fills pores more effectively. This resulted in a denser stabilised composite. SEM–EDX results further confirmed increased Ca, and comparatively lower Si and Al, in the M4 mixes, which yielded more calcite and less CSH and CASH, respectively, thereby densifying these mixes but with slightly lower UCSs, precipitating more insoluble metal complexes, and consequently decreasing the concentrations of heavy metals—all confirming further the TCLP, UCS, and XRD results.

In summary, it is evident that NCOS reinforced and improved the grading of poorly graded MWR, and that COSP stabilised and immobilised the heavy metals present in the MWR, thereby resulting in a two-waste composite with improved strength and other engineering properties that makes it suitable for road pavement subgrade use.

## Figures and Tables

**Figure 1 materials-15-02916-f001:**
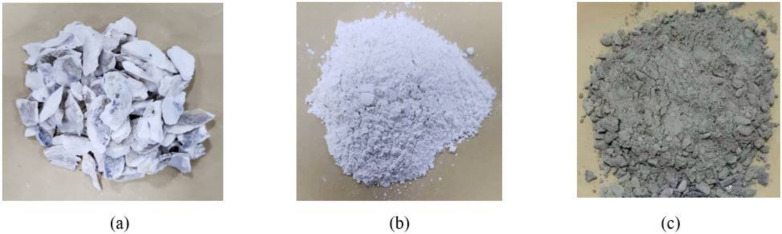
Materials: (**a**) NCOS (10–13 mm); (**b**) COSP; (**c**) MWR (≤5 mm).

**Figure 2 materials-15-02916-f002:**
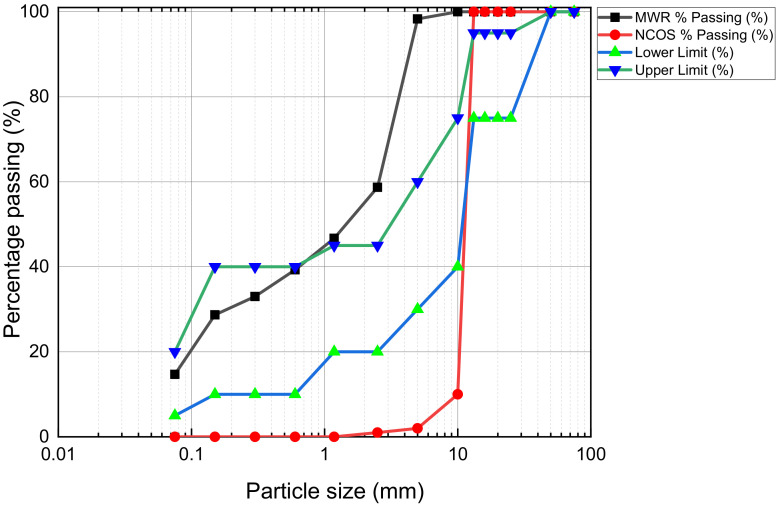
Particle size distribution of mine waste rock and NCOS used in laboratory tests/experiments.

**Figure 3 materials-15-02916-f003:**
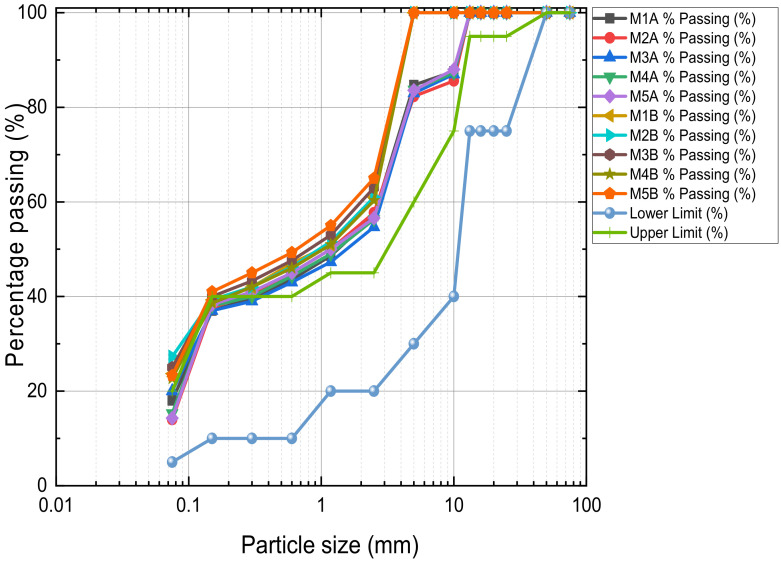
Particle size distribution of the ten (10) design mixes.

**Figure 4 materials-15-02916-f004:**
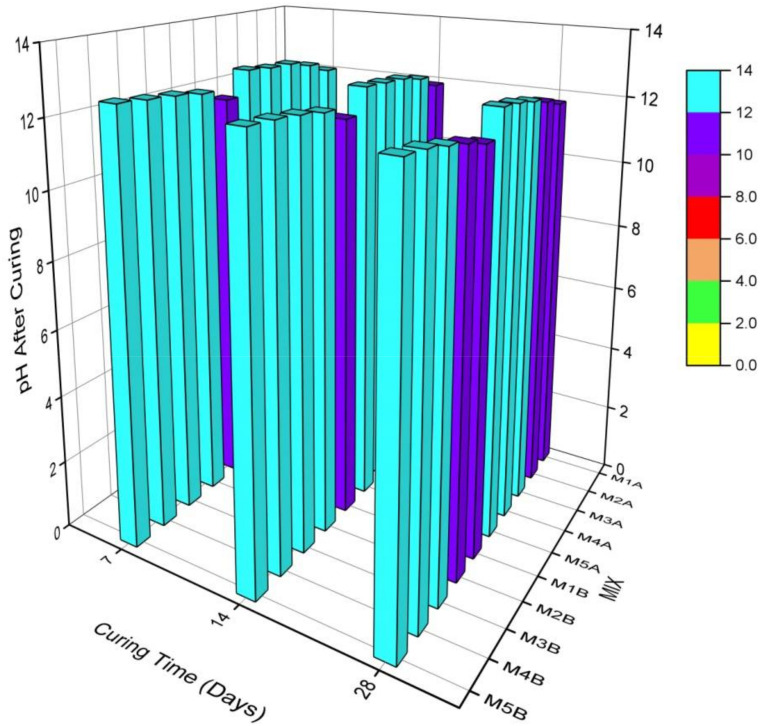
pH after curing the different mixes of stabilised MWR at different curing ages.

**Figure 5 materials-15-02916-f005:**
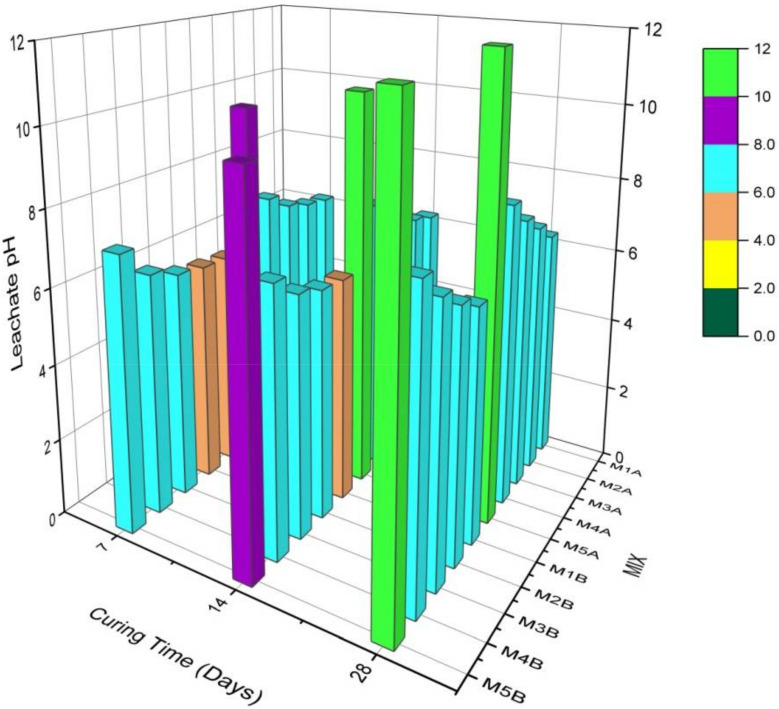
Leachate pH of the different mixes of stabilised MWR at different curing ages.

**Figure 6 materials-15-02916-f006:**
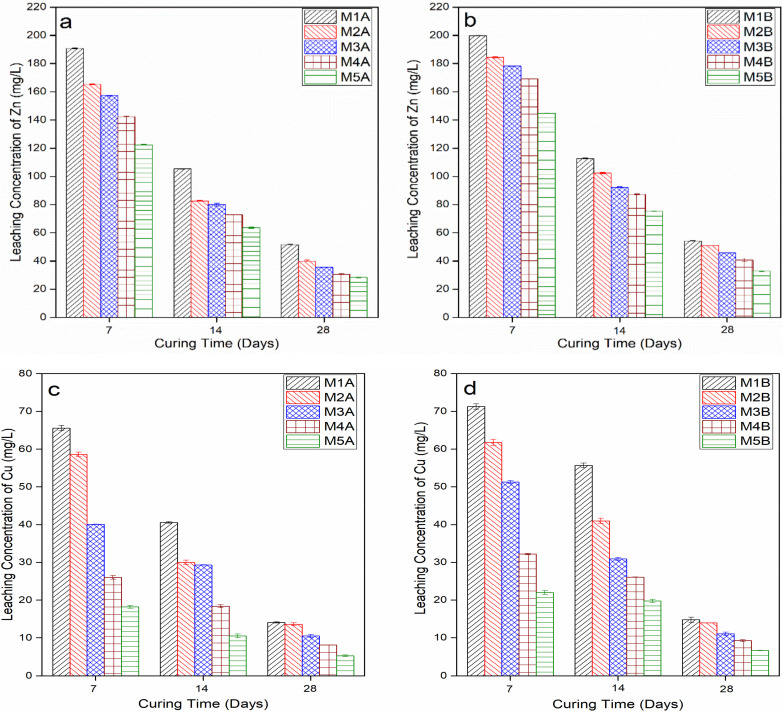
Leaching concentration vs. curing time: (**a**) leachate concentration of Zn in A samples, (**b**) leachate concentration of Zn in B samples, (**c**) leachate concentration of Cu in A samples, (**d**) leachate concentration of Cu in B samples, (**e**) leachate concentration of Pb in A samples, (**f**) leachate concentration of Pb in B samples.

**Figure 7 materials-15-02916-f007:**
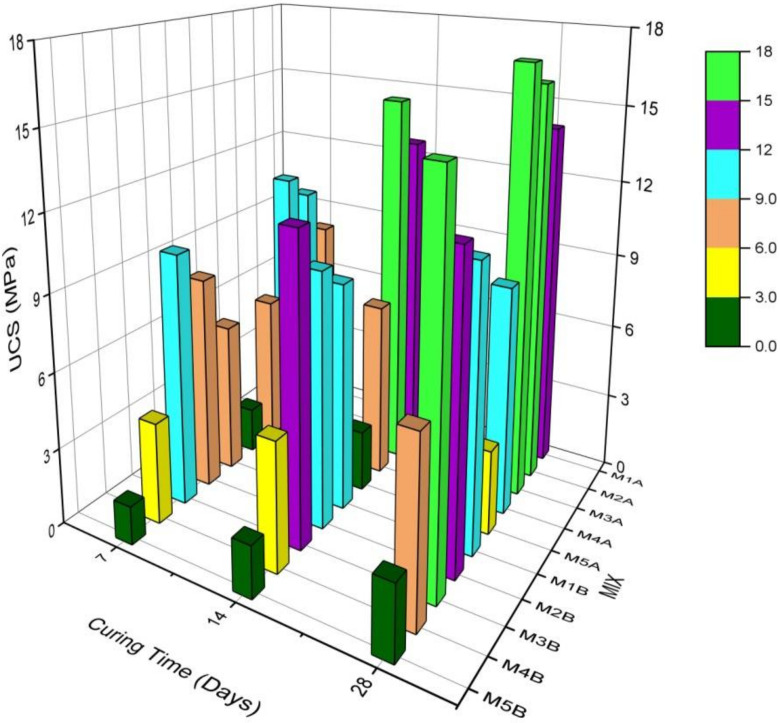
Unsoaked UCS of different mixes of stabilised MWR at different curing ages.

**Figure 8 materials-15-02916-f008:**
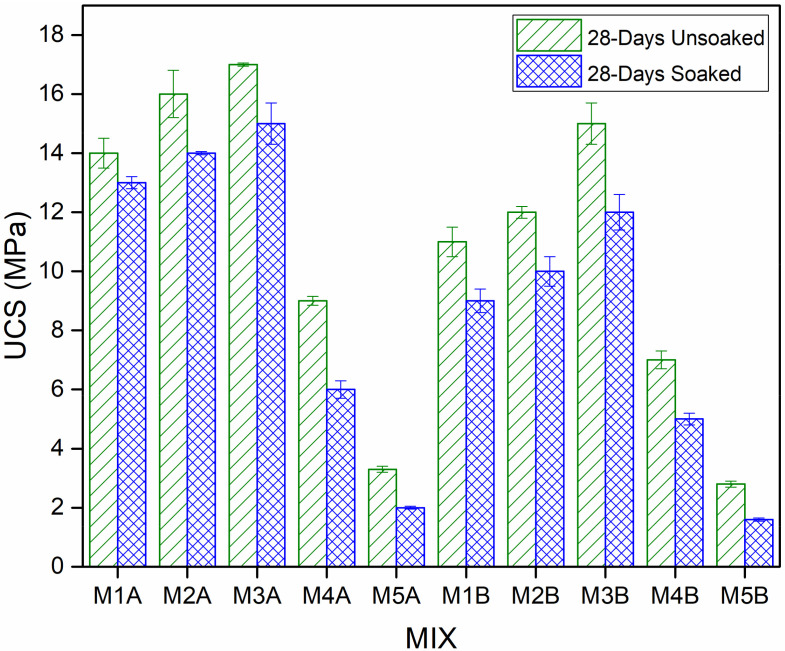
Unsoaked 28-day UCS and soaked 28-day UCS of different mixes of stabilised MWR.

**Figure 9 materials-15-02916-f009:**
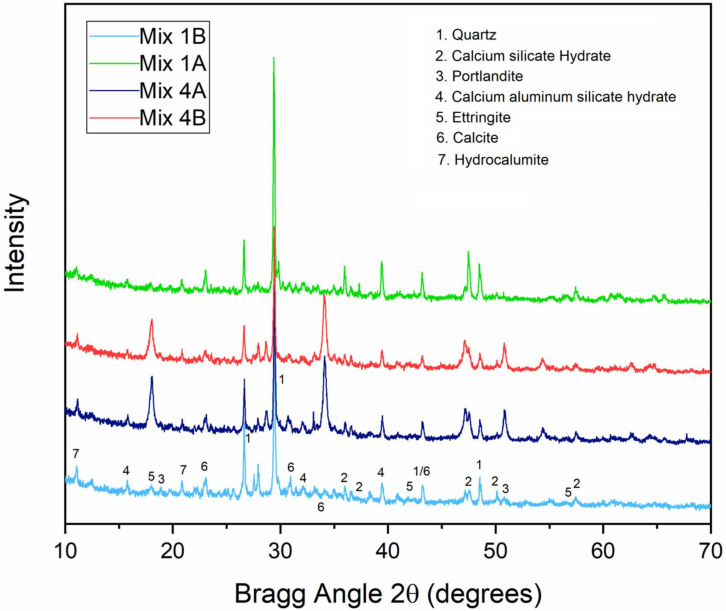
XRD diffractogram of M1A, M1B, M4A, and M4B of stabilised MWR at 28 days.

**Figure 10 materials-15-02916-f010:**
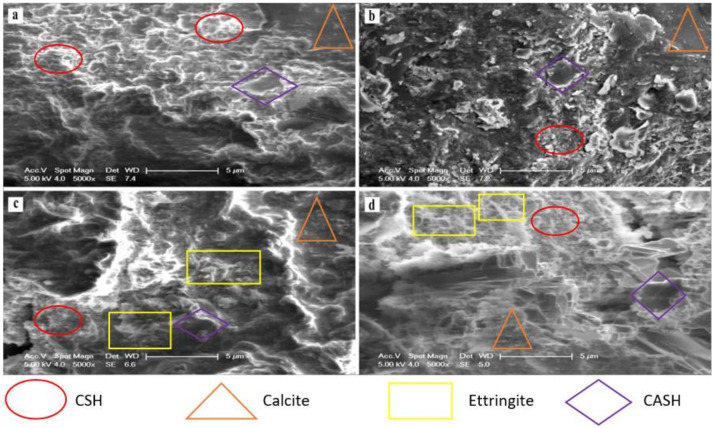
SEM micrographs of (**a**) M1A, (**b**) M1B, (**c**) M4A, and (**d**) M4B stabilised MWR at 28-day curing age.

**Figure 11 materials-15-02916-f011:**
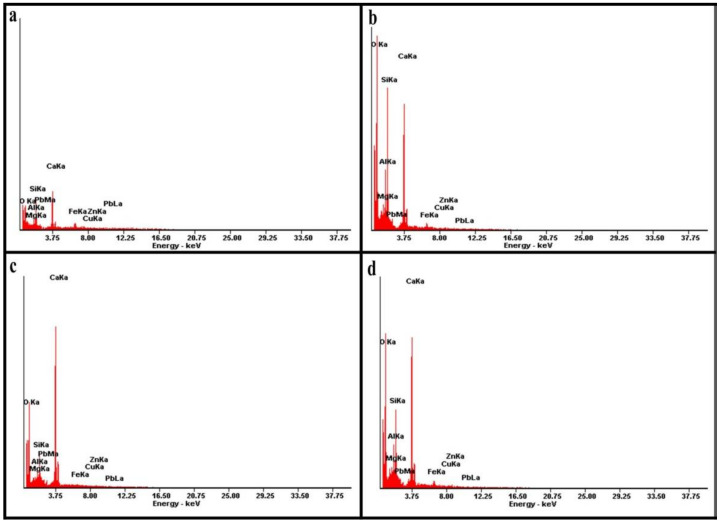
SEM–EDX spectrographs (**a**) M1A, (**b**) M1B, (**c**) M4A, and (**d**) M4B stabilised MWR at 28-day curing age.

**Table 1 materials-15-02916-t001:** Physical properties and chemical compositions of materials.

Property	Index	MWR	NCOS	OPC	COSP
Physical properties	Natural moisture content (%)	3.49	/	/	/
	Specific gravity, G_s_	2.62	/	3.14	2.39
	Liquid limit (%)	34.74	/	/	/
	Plastic limit (%)	21.85	/	/	/
	Plasticity index	12.9	/	/	/
	Gravel fraction: 2–5 mm (%)	43.00	/	/	/
	Sand fraction: 0.06–2 mm (%)	47.00	/	/	/
	Silt and Clay fraction: <0.06 mm (%)	10.00	/	/	/
	Soil classification: Poorly-graded gravelly sand	GP	/	/	/
	Unconfined compressive strength (Mpa)	0.13	/	/	/
	pH	8.70	9.60	12.30	12.70
	Water absorption (%)	/	2.10	/	/
	Organic matter (%)	/	1.57	/	/
	Size (mm)	≤5	10–13	≈0.04	≈0.01
Chemical compositions (Wt. %)	Calcium oxide (CaO)	56.7	95.54	59.80	97.50
	Silicon oxide (SiO_2_)	17.9	0.25	22.31	0.21
	Aluminium oxide (Al_2_O_3_)	7.08	0.90	6.29	0.07
	Magnesium oxide (MgO)	1.50	0.42	3.40	0.42
	Potassium oxide (K_2_O)	2.02	0.01	0.70	0.02
	Sulfate oxide (SO_3_)	0.02	0.60	4.01	0.02
	Sodium oxide (Na_2_O)	/	0.36	0.69	0.13
	Ferric oxide (Fe_2_O_3_)	14.6	1.10	2.55	0.15
	Chloride (Cl)	/	0.74	/	0.11
	Others	0.05	/	/	/
	Loss of ignition	0.06	0.08	0.25	1.19

**Table 2 materials-15-02916-t002:** Mix design.

Design Mix	Mix 1A	Mix 2A	Mix 3A	Mix 4A	Mix 5A	Mix 1B	Mix 2B	Mix 3B	Mix 4B	Mix 5B
Cement (% by total Wt.)	20	15	10	5	0	20	15	10	5	0
COSP (% by total Wt.)	0	5	10	15	20	0	5	10	15	20
NCOS (% by total Wt.)	15	15	15	15	15	0	0	0	0	0

**Table 3 materials-15-02916-t003:** SEM–EDX elemental atomic percentages.

Element	Mix 1A Atomic %	Mix 4A Atomic %	Mix 1B Atomic %	Mix 4B Atomic %
O	65.25	69.82	67.32	70.49
Mg	2.42	1.39	2.63	1.47
Al	3.67	2.01	4.42	3.71
Si	8.71	3.07	9.91	6.14
Ca	16.68	21.46	10.16	14.76
Fe	2.32	1.53	3.03	2.24
Cu	0.24	0.2	0.72	0.24
Zn	0.43	0.37	0.86	0.47
Pb	0.28	0.15	0.95	0.48

## Data Availability

Data is contained within the article.

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
