# Peer review of "Mechanical, Leaching, and Microstructure Properties of Mine Waste Rock Reinforced and Stabilised with Waste Oyster Shell for Road Subgrade Use"

_materials, 2022, doi:10.3390/ma15082916_

Round 1

Reviewer 1 Report

Overall, the work has several limitations. The introduction is very repetitive. The naming of properties and/or techniques is incorrect. Taking into account the defined MWR particle size (<=5mm) Figure 1 is not very noticeable. Partcle size up to 100 mm and scale only up to 100%. Why didn't they measure the pH of NCOS (maybe around 9 or bigger)? MWR contamination protocol ? quantities ? Why nitrates if they are relatively uncommon in a mine environment and are considered inert to cement hydratation by the authors? Severe leaching conditions? XRD and SEM acquisition conditions - very important for data interpretation (eg Cu source or Ni?) why to start at 10 if you need to check the formation of ettrinite or other hydrated products? Some differences (e.g. pH after curing in MA and MB tests are very narrow! Analysis of leaching (pH and contents) requires initial values ​​to serve as a control. The interpretations may be more accurate, considering the role of Ca(OH)2 /cement. Many aspects are not observed in the tests, only deduced. The functions of cement, portlandite, calcite, as sources of calcium are not clearly distinguished. The particle size of COSP and cement is not presented, so this aspect is not valid to justify UCS differences. Figure 5 is not good at showing all UCS data. The XRD analysis is very limited and apparently contains many errors in the marking of angles and respective phases. For example MIX1 has more quartz because the original material has no added NCOS calcite. On the other hand MIX 1A has a lot of calcite as it represents the final evolution of several products and because it originally contains 15% calcite. SEM analysis has limitations in terms of interpretation and image quality (very bad). To make considerations about the contaminants a dedicated analysis by EDX is necessary. Additional images on microstructure may eventually help to understand the improvements found in some proposed mixtures.

Author Response

Cover Letter to Reviewer 1

(Please note that all ‘changes’ or ‘sections with changes’ are coloured ‘red’ in the revised manuscript, due to the fact that the revision was extensive, ‘track changes’ would have been a bit ‘untidy’)

The introduction is very repetitive.

INTRODUCTION revised; eliminating repetitions. 

The naming of properties and/or techniques is incorrect.

Properties and/or techniques nomenclature reviewed throughout.

Taking into account the defined MWR particle size (≤ 5mm) Figure 1 is not very noticeable. Particle size up to 100 mm and scale only up to 100%.

Grading envelope that gives ‘Lower and Upper Limits’ for suitable subgrade material is used to show how deficient/unsuitable the ≤ 5mm MWR is as road pavement subgrade material. Grading limits are normally up to 75mm and since the logarithmic scale is to ‘base 10’, hence the reason for the last particle size on the graph being 100mm. My plot actually ended at 75mm, as a result of the specified grading limits.

Also, since % passing is in % i.e., over 100; this is why the scale is limited to100%. The particle size distribution of the unstabilised MWR is outside the grading envelope; implying it is unsuitable for road pavement subgrade material - the graph was intentionally plotted to highlight the foregoing. Therefore, suggesting that the ≤ 5mm MWR needs to be reinforced and/or stabilised; underpinning one of the justifications for the study undertaken in my research. 

Why didn't they measure the pH of NCOS (maybe around 9 or bigger)?

I have included the pH of NCOS in Table 1. Its value being 9.60.

MWR contamination protocol? quantities?

Chinese MEP states that Zinc (Zn), Copper (Cu) and Lead (Pb) concentrations in soils should satisfy the following: Zn ≤ 300mg/kg or ≤ 100mg/L, Cu ≤ 200mg/kg or ≤ 15mg/L and Pb ≤ 80mg/kg or ≤ 5mg/L. MWR was spiked 15 times above contamination levels with the three (3) heavy metals, i.e., 4500mg/kg with Zn, 3000mg/kg with Cu and 1200mg/kg with Pb. I have included this information in the manuscript.

Why nitrates if they are relatively uncommon in a mine environment and are considered inert to cement hydratation by the authors?

Raw/pure metals, i.e. pure Zn, Cu and Pb were not available in the desired quantities. Their various salts (chlorides, nitrates etc.) were available in the quantities required. ‘ Du, Y-J.; Jiang, N-J.; Liu, S-Y.; Jin, F.; Singh, D.; Puppala, A.J. Engineering properties and microstructural characteristics of cement-stabilized zinc-contaminated kaolin. Can. Geotech. J. 2014, 51, 289–302, doi:10.1139/cgj-2013-0177’ in their study stated that nitrates are inert to cement hydration, so it would not adversely affect the engineering properties of the stabilised soil. Implying, that the same effect will be experienced if raw/pure metals were

used to spike the soil. This was why I opted to use the nitrates salts of the metals in the absence of their raw/pure alternatives. I used the molecular weight of the nitrate salts, the molecular weight of the pure metals and the desired concentrations to get the actual quantities for spiking.

Severe leaching conditions?

TCLP was performed according to EPA TCLP method 1311 (US EPA, 1992). I have noted it in the manuscript.

XRD and SEM acquisition conditions - very important for data interpretation (eg Cu source or Ni?) why to start at 10 if you need to check the formation of ettringite or other hydrated products?

Crushed 28-days UCS samples were used for both XRD and SEM analysis. For XRD analysis, samples were sieved through a 0.075 mm sieve to get a fine powder, and were scanned in ranges from 0° to 90° (2ÆŸ) using a Rigaku D/Max-2500 X-ray diffractometer with a Ni-filtered Cu-Kα radiation source at 40 kV and 30 mA, and at 0.02° intervals and a counting time of 1s per step in order to identify the crystalline phases. However, data between ranges 10° to 70° was recorded because significant peaks were detected within this range. No peaks were detected below 10° and above 70°. Jade software version 7.1 (MDI 2005) was used to conduct the qualitative analyses of the XRD patterns, using the patterns present in the International Centre for Diffraction Data database (ICDD 2002) as reference. For SEM analysis, SEM SU8010, Hitachi, Japan equipped with an energy dispersive X-ray (EDX) spectroscope was employed to acquire micrographs at different resolutions and magnifications and the chemical compositions of the stabilised samples. Samples were placed on aluminum stubs using double-sided carbon tape prior to SEM analysis, and were then placed in the machine for image capture at 5kV. I have included this information in the manuscript.

Some differences (e.g. pH after curing in MA and MB tests are very narrow!

This is due to the fact that the only difference between A mixes and their corresponding B counterparts is 15% NCOS which is relatively small and NCOS is also not in powdered form. Therefore, significant pH/other differences are not observed in some cases.

Analysis of leaching (pH and contents) requires initial values ​​to serve as a control.

MWR was spiked 15 times above contamination levels with the three (3) heavy metals, i.e., 4500mg/kg with Zn, 3000mg/kg with Cu and 1200mg/kg with Pb. The initial heavy metal concentrations of the spiked MWR are as follows: Zn = 216mg/L, Cu = 84mg/L and Pb = 13.2mg/L. I have included them in the manuscript.

The interpretations may be more accurate, considering the role of Ca(OH)2 /cement.

In the analysis and discussions, I emphasised the effect of the various proportions of cement, COSP and NCOS on all properties studied in the research. COSP and NCOS provides significant Ca(OH)2 and the effect this increased calcium has, on all properties studied in the research was discussed.

Many aspects are not observed in the tests, only deduced.

I was concerned about ending up with a lengthy manuscript, but have now included the observed aspects of the tests in the manuscript as suggested. A main portion of the discuss is noted under ‘2.2.1. Mechanism of Stabilisation and Immobilisation of Heavy Metal Contaminated Soils’. Other portions are captured in as ‘changes’ coloured in ‘red’ in the manuscript.

The functions of cement, portlandite, calcite, as sources of calcium are not clearly distinguished.

Their functions have been discussed under the various properties and subsequent analysis.

The particle size of COSP and cement is not presented, so this aspect is not valid to justify UCS differences.

The particle size of COSP and OPC have now been included in Table 1. COSP ≈ 0.01mm and OPC ≈ 0.04mm.

Figure 5 is not good at showing all UCS data.

I have re-plotted the 3-D bar graphs to show all the data. Changing; the orientation and sequence of the values. However, M1A – 14days is still not showing. Any further twisting or alteration of the graph for it to show, will affect other data. Apparently, that is the best angle of the graph showing all but one data. The trends UCS values follow in A samples and B samples and as cement content decrease and COSP content increase are stated in the analysis and will help in the absence of M1A – 14days on the graph.

The XRD analysis is very limited and apparently contains many errors in the marking of angles and respective phases. For example, MIX 1 has more quartz because the original material has no added NCOS calcite. On the other hand, MIX 1A has a lot of calcite as it represents the final evolution of several products and because it originally contains 15% calcite.

I have rechecked the plotting of angles and respective phases; they are correct and represent the results obtained and analysed. I have tried to explain the XRD results and analysis in a better way, as can be seen done in ‘red’ under section ‘3.3’.

SEM analysis has limitations in terms of interpretation and image quality (very bad).

I have interpreted and explained the SEM results and analysis, as can be seen in ‘red’; done with an improved image quality as suggested.

To make considerations about the contaminants a dedicated analysis by EDX is necessary. 

I have included the EDX results and spectrographs that go with the results. Subsequent analysis and discussions have also been done. SEM–EDX spectrograghs and elemental atomic percentages are now shown in Fig. 10 and Table 3 respectively.

Additional images on microstructure may eventually help to understand the improvements found in some proposed mixtures.

In-depth analysis to help understand the improvements found in the proposed mixtures have now been done. The 5µm micrographs gave the best images.

Reviewer 2 Report

The research approaches the usage of mining waste as subgrade material for pavements. The poor grade of the waste has been improved using crushed oyster shells as coarse aggregate, and the presence of heavy metals has been counteracted by replacing partially cement with calcined oyster shells powder.  

The work is well structured, the experimental design is appropriate, and the conclusions are relevant. However, some aspects can be improved concerning the description of the methods used and the presentation of results:

  1. References should be included in the text in a different way. For instance, in line 39, “… as highlighted by [3,4]” would be better in the form “… as highlighted by several authors [3,4]”, or in line 83 “[21] affirms…” would be better “Kumpiene et al. [21] affirm….”
  2. There is a lack of photographs in the manuscript. Pictures are welcome to document the research process. If available, it would be desirable to include photos of the used materials, the samples, and the performed tests.
  3. Figure 1 is not clear enough. There should be shown the particle size distributions of the mining waste and the oyster shell aggregate separately. And on the other hand, perhaps in another figure, the grading curves of the different mixes.
  4. Figures must be located in the text within the paragraph they are referred to. For instance, figures 2 and 3 should be included in paragraph 3.1.1
  5. The discussion paragraph is well-argued but needs comparison with the results obtained by other authors with the same or similar materials to solve the problem of stabilising heavy metals in mine waste aggregates.
  6. In line 331, the statement “It should also be noted that considerable shrinkage occurs in cement-only mixes” should be supported by references, given that shrinkage has not been studied in this work.
  7. The paragraph of the conclusions is too long. It must be shortened by removing the details previously exposed in the discussion.
  8. A last conclusion should be added summarising the overall finding of the research, similar to that exposed in the abstract (lines 17-20)

Author Response

Cover Letter to Reviewer 2

(Please note that all ‘changes’ or ‘sections with changes’ are coloured ‘purple’ in the revised manuscript, due to the fact that the revision was extensive, ‘track changes’ would have been a bit ‘untidy’)

  1. References should be included in the text in a different way. For instance, in line 39, “… as highlighted by [3,4]” would be better in the form “… as highlighted by several authors [3,4]”, or in line 83 “[21] affirms…” would be better “Kumpiene et al. [21] affirm….”

       In-text references have been done as suggested.

  1. There is a lack of photographs in the manuscript. Pictures are welcome to document the research process. If available, it would be desirable to include photos of the used materials, the samples, and the performed tests.

I have included photographs of the samples used. Being that I did not see pictures of samples and performed tests in published articles read, I thought photographs should not be included therefore, I did not take such pictures during the laboratory tests. I will pictorially document tests and processes in future.

  1. Figure 1 is not clear enough. There should be shown the particle size distributions of the mining waste and the oyster shell aggregate separately. And on the other hand, perhaps in another figure, the grading curves of the different mixes.

I have re-plotted Figure 1 to effect the suggested changes. I have also done another figure, i.e. Figure 2 to show the grading curves of the ten (10) different mixes.

  1. Figures must be located in the text within the paragraph they are referred to. For instance, figures 2 and 3 should be included in paragraph 3.1.1

I tried doing them as suggested, but some figures would not fit in the remaining spaces that where available. And if I move on to the next page to insert the figures, the space that was not filled remained blank and the manuscript ran into much more pages. So, I have left them as they are, in order to maximize the usage of all available space. But if length of manuscript or number of pages is not an issue, I will do as suggested. Please advise.

  1. The discussion paragraph is well-argued but needs comparison with the results obtained by other authors with the same or similar materials to solve the problem of stabilising heavy metals in mine waste aggregates.

I have now done more comparison with, or cited more authors consistent with the highlighted concern.

  1. In line 331, the statement “It should also be noted that considerable shrinkage occurs in cement-only mixes” should be supported by references, given that shrinkage has not been studied in this work.

I have now supported the statement with references.

  1. The paragraph of the conclusions is too long. It must be shortened by removing the details previously exposed in the discussion.

I tried editing/shortening the ‘Conclusions’, I was only able to delete about three (3) or four (4) lines that have already been captured in the ‘Discussions’. Doing so distorted the meanings of the paragraphs they were in. I am kindly asking permission to leave the ‘Conclusions’ as it is, as it is concise and also contains data not present in the ‘ Discussion’.

  1. A last conclusion should be added summarizing the overall finding of the research, similar to that exposed in the abstract (lines 17-20)

I have done a summary of the ‘conclusions’ summarizing the overall findings of the research.

Reviewer 3 Report

The research objective is worth investigation as the oyster shell is a renewable source of materials. The manuscript provides sufficient details on the background, test materials, test methods, and test results. With that, the following comments need to be addressed before it can be considered for publication:

  1. There are some grammar and format errors. Some are listed below:

Line 33, in “Citing [1,2];”, the semicolon should be replaced with a comma. This type error appears in many other places (e.g. , Lines 178, 192, 195).

Line 47, “confirms” should be “confirm”. This type of error also occurred in many other places (e.g., lines 75, 77, 178, 182, 197, 214, 233, 250, 251).

Line 58, should “same” be “some”?

Line 70 and Line 86, these two sentences are essentially the same. Please delete one.

Lines 93-95, Is this statement from this study or from the literature? If from this study, please delete it because you have not introduced the study yet while giving the conclusions. If from the literature, please add citations to the references.

Line 99, consider replacing “than” with “but not” or “rather than”.

Lines 105, 107, 111 and many others, add a space between a number and its unit.

In Figs 2 and 3, do not use smooth curves to represent the gradations in Figure 2. Use lines that connect all points in a gradation.

Line 288, change “Fig.” to “Figs.”

  1. What is the number of replicates in each test? The bar charts do not show the level of variability in the test results. It is suggest that error bars be added to the charts.

  1. Discussion of the test results was mainly based on visual assessment, which is less convincing. It is suggested that statistical testing such as t-test or ANOVA should be performed.

  1. Other material properties such as coefficient of thermal expansion may significantly affect the performance of pavement base/subgrade. It is suggested that the authors also evaluate other relevant properties of their materials.

Author Response

(Please note that all ‘changes’ or ‘sections with changes’ are coloured ‘red’ in the revised manuscript)

Let me firstly express my profound thanks and appreciation for your comments/suggestions which has added to the overall value of my work. Below are the responses to your input:

  • There are some grammar and format errors. Some are listed below:

All citations and other grammatical and formatting errors have been corrected accordingly; repetitions have also been deleted.

  • Line 58, should “same” be “some”?

“It is advisable to excavate such soils and replace same with suitable borrow pit gravel as using such in-situ soils will require very costly measures to improve their engineering properties.”

It should read as “same”, meaning that unsuitable soils should be replaced with suitable borrow pit gravel.

  • Lines 105, 107, 111 and many others, add a space between a number and its unit.

Spaces have been added between all numbers and their respective units.

  • In Figs 2 and 3, do not use smooth curves to represent the gradations in Figure 2. Use lines that connect all points in a gradation.

Figs. 2 and 3 have been re-plotted with lines connecting all points as directed.

  • What is the number of replicates in each test? The bar charts do not show the level of variability in the test results. It is suggested that error bars be added to the charts.

All tests/experiments were done in duplicates and average values were reported only if the individual measurements were within an error of 10% as suggested by Deok Hyun Moon et al. 2013 and 2015. The results showed that the values of the standard deviation were less than 5% for the duplicates, demonstrating the reproducibility of the test results. Bar charts have been re-plotted to show the level of variability in the test results by adding “error bars” to the charts.

  • Discussion of the test results was mainly based on visual assessment, which is less convincing. It is suggested that statistical testing such as t-test or ANOVA should be performed.

As suggested, statistical analysis using Minitab 21 One-way ANOVA has been performed to assess the performance of the different mixes, whereby P Ë‚ 0.05 was considered statistically significant. Statistical analysis using Minitab 21 One-way ANOVA confirmed that the results were statistically significant for all curing periods, with P values  Ë‚ 0.05 (P = 0.000). Below is a summary of the ‘ Anova output’ for the UCS results at the different curing ages:

7 Days UCS

Analysis of Variance

Source

DF

Adj SS

Adj MS

F-Value

P-Value

Mix

9

211.600

23.5112

101.12

0.000

Error

10

2.325

0.2325

Total

19

213.925

Model Summary

S

R-sq

R-sq(adj)

R-sq(pred)

0.482183

98.91%

97.94%

95.65%

14 Days UCS

Analysis of Variance

Source

DF

Adj SS

Adj MS

F-Value

P-Value

Mix

9

350.925

38.9916

115.19

0.000

Error

10

3.385

0.3385

Total

19

354.310

Model Summary

S

R-sq

R-sq(adj)

R-sq(pred)

0.581808

99.04%

98.18%

96.18%

28 Days UCS

Analysis of Variance

Source

DF

Adj SS

Adj MS

F-Value

P-Value

Mix

9

466.302

51.8113

143.52

0.000

Error

10

3.610

0.3610

Total

19

469.912

Model Summary

S

R-sq

R-sq(adj)

R-sq(pred)

0.600833

99.23%

98.54%

96.93%

All the P values are 0.000. I am therefore asking your kind permission that I just ‘mention’ the results and its significance in the main body of the manuscript instead of including them (done so in the manuscript). As this will cause the manuscript to run into a lot more pages. P = 0.000 is statistically significant and confirms all my results, visual assessments and discussions.

  • Other material properties such as coefficient of thermal expansion may significantly affect the performance of pavement base/subgrade. It is suggested that the authors also evaluate other relevant properties of their materials.

The scope of the study considered the UCS, leachability and microstructure of the cement-waste oyster shell (WOS) stabilised composites. Therefore, only material properties directly related to the aforementioned parameters were investigated and stated. Notwithstanding, over the years, crushed oyster shells have been used as base and subbase material and according to E. Doran Jr. [23] with durability offering lifespans of up to 18 years and optimal mixes meeting Texas Highway Department specifications for Class I flexible base. 

Reviewer 4 Report

The authors made all necessary corrections. I agree to accept the article

Author Response

My profound thanks and appreciation for your comments/suggestions which has added to the overall value of my work. Thanks  for accepting the article.

Round 2

Reviewer 1 Report

Despite the new version having considered some suggestions proposed by the reviewers, some critical aspects were not reassessed. It is verified that the analysis by XRD and SEM, presents many deficiencies, graphical and of interpretation. The authors continue, for example, to indicate that quartz has a reflection around 30º. SEM images are very deficient and do not allow for the minerochemical considerations presented by the authors. The identification of mineral phases by without requires chemical and morphological validation of the crystals. For example ettringite is very easily recognizable and in these images it is impossible to see its typical acicular shape. When discussing the results, it is important to clarify the aspect of replacing cement with lime, as the retention of metals and the physical properties are strongly dependent on these products. The mechanical part is also affected by the global texture of the aggregates. In summary, I think that points 3.3 and 3.4 really need a significant change, so that the conclusions are well founded.

Author Response

Cover Letter Reviewer 1
(Please note that all ‘changes’ or ‘sections with changes’ are coloured ‘red’ in the revised manuscript, due to the fact that the revisions were extensive, ‘track changes’ would have been a bit ‘untidy’)

Comment:
Despite the new version having considered some suggestions proposed by the reviewers, some critical aspects were not reassessed. It is verified that the analysis by XRD and SEM, presents many deficiencies, graphical and of interpretation. The authors continue, for example, to indicate that quartz has a reflection around 30º. 

From my results and plots, Quartz actually peaked at 2ÆŸ = 29° but decided to round up. Evidently, Quartz from laterite, mud etc, normally peak around 26°- 27° notwithstanding, being that MWR is not lateritic but mostly made up of silica; hence the difference in the value of 2ÆŸ. Furthermore, MWR is weathered/crushed rock, with higher amount of silica. The origin and composition of MWR, the combination and quantities of materials and heavy metals mixed, cured and tested  - all of which affected the 2ÆŸ of Quartz.
Supplementary pictures of Quartz peaking at angles greater than the normal 26°- 27° will be provided with this ‘cover letter’ in order to show that Quartz in some cases, peaks at 2ÆŸ angles other than the usual  26°- 27°. The same also applies to some of the other stabilisation products like Calcite. Photographic evidence are also provided. 

Comment:
SEM images are very deficient and do not allow for the minerochemical considerations presented by the authors. The identification of mineral phases by without requires chemical and morphological validation of the crystals. For example ettringite is very easily recognizable and in these images it is impossible to see its typical acicular shape. 

The typical needle-like ettringite, gel-like CSH, platy CASH, and bulky calcite could be seen in their characteristic forms on all four micrographs of Fig. 11 (as highlighted) and can be compared and confirmed from the micrographs and reports of the following authors [21, 89, 90, 91] as referenced in the manuscript, also obtained similar images in their studies. 

Comment:
When discussing the results, it is important to clarify the aspect of replacing cement with lime, as the retention of metals and the physical properties are strongly dependent on these products. The mechanical part is also affected by the global texture of the aggregates.

I have further elaborated on the aspect of replacing cement with lime (COSP) and the mechanical aspect of the results obtained; indicating how they are linked to the design mix (proportions of OPC and COSP) and other properties/results. Replacing OPC with COSP causes less CSH and CASH to be formed which affects/lowers UCS. Consequently also, COSP increases the pH of the stabilised composite, which causes heavy metals to be precipitated at such high pHs, increased COSP contents further causes the increased formation of Calcite and portlandite which further precipitates insoluble heavy metal complexes thus significantly lowering the heavy metal concentrations of high COSP mixes as compared to OPC mixes. The insoluble heavy metal complexes that caused heavy metal retention have been included in the manuscript in section 3.3. 

Comment:
In summary, I think that points 3.3 and 3.4 really need a significant change, so that the conclusions are well founded.

I have further explained the XRD and SEM results (section 3.3 and 3.4) in conjunction with the TCLP, UCS and SEM-EDX results and subsequently improved on the ‘Conclusions’ made, so that they are well founded and supported by the ‘Results and Discussions’. Certain aspects can be directly deduced from each property/test/experiment whilst others are implied or can be confirmed through another property/test/experiment. For e.g.; the UCS of M4A is less than the UCS of M1A whilst the heavy metal concentrations of M4A are less than those of M1A. This means that there is something in M1A that causes increased strength but more heavy metal concentration and there is something in M4A that is responsible for decreased strength and lower heavy metal concentrations. The first thing that is looked into is the differences in the mix design, one (M1A) has more OPC and the other (M4A) has more COSP. 
Past literature as well as my results have shown that mixes with more OPC have greater UCS than those with more lime (COSP). Furthermore, from past literature as well as my results, it is evident that mixes with more lime (COSP), have high alkalinity and normally yields lower heavy metal concentrations than those with little or no lime (COSP). So, we now go to the XRD and SEM-EDX results to see which stabilisation products are exactly responsible for increased UCS in M1 and decreased heavy metal concentrations in M4. XRD results showed that M1 mixes had more CSH and CASH and less ettringite and calcite than M4 mixes. This automatically tells us that CSH and CASH in M1 mixes are responsible for increased UCS and the lower ettringite and calcite are responsible for relatively higher heavy metal concentrations (due to relatively lower alkalinity). The reverse occurs in M4. In the XRD diffractograms, peaks of the insoluble metal complexes mostly overlapped with those of the stabilisation products, so they were not included in the XRD plot, as it would have caused the plot to be very untidy. This is acceptable as the formation and presence of the insoluble heavy metal complexes are confirmed in the SEM-EDX results and images.
Reinforcing the foregoing; the SEM-EDX results now comes in as a confirmatory measure; endorsing the presence and quantities of the aforementioned products and the reasons for the results obtained and trends observed in the whole study. They even further confirmed the low heavy metal concentrations of the three (3) metals obtained from TCLP results, confirming they were stabilised and immobilised by the mentioned products. So, when TCLP, UCS, XRD and SEM-EDX results are analysed in unison the whole concept is clearly visualised and established.

Note of Appreciation:

I wish to take this opportunity to thank you for your vital input which in so many ways has added great value to my research. 

Respectfully Yours,
Nadia Wurie.
